# Predicting physician departure with machine learning on EHR use patterns: A longitudinal cohort from a large multi-specialty ambulatory practice

Kevin Lopez[1], Huan Li[1,2], Hyung Paek[3,4], Brian Williams[4], Bidisha Nath[1], Edward R. Melnick[1,5]‡*, Andrew J. Loza[1]‡

1 Department of Emergency Medicine, Yale School of Medicine, New Haven, Connecticut, United States of America, 2 Computational Biology and Bioinformatics, Yale School of Medicine, New Haven, Connecticut, United States of America, 3 Information Technology Services, Yale New Haven Health, Stratford, Connecticut, United States of America, 4 Northeast Medical Group, Yale New Haven Health, New London, Connecticut, United States of America, 5 Department of Biostatistics (Health Informatics), Yale School of Public Health, New Haven, Connecticut, United States of America

☯ These authors contributed equally to this work.
‡ ERM and AJL are joint senior authors.
* edward.melnick@yale.edu

**Data Availability Statement:** Data cannot be shared publicly because of concerns for participant

## Abstract

Physician turnover places a heavy burden on the healthcare industry, patients, physicians, and their families. Having a mechanism in place to identify physicians at risk for departure could help target appropriate interventions that prevent departure. We have collected physician characteristics, electronic health record (EHR) use patterns, and clinical productivity data from a large ambulatory based practice of non-teaching physicians to build a predictive model. We use several techniques to identify possible intervenable variables. Specifically, we used gradient boosted trees to predict the probability of a physician departing within an interval of 6 months. Several variables significantly contributed to predicting physician departure including *tenure (time since hiring date)*, *panel complexity*, *physician demand*, *physician age*, *inbox*, *and documentation time*. These variables were identified by training, validating, and testing the model followed by computing SHAP (SHapley Additive exPlanation) values to investigate which variables influence the model's prediction the most. We found these top variables to have large interactions with other variables indicating their importance. Since these variables may be predictive of physician departure, they could prove useful to identify at risk physicians such who would benefit from targeted interventions.

## Introduction

Physician turnover disrupts continuity of care for patients, affects clinical outcomes [1–3] and imposes a significant financial burden on the healthcare industry [4]. With each physician

employment. The code is available on github at
https://github.com/kevkid/physician_retention_2.0.

**Funding:** This work was supported by an American
Medical Association (AMA) Practice
Transformation Initiative (contract No 16118). The
funders had no role in study design, data collection
and analysis, decision to publish, or preparation of
the manuscript. Dr. Melnick reports receiving
grants from the National Institute on Drug Abuse
and Agency for Healthcare Research and Quality
outside the submitted work.

**Competing interests:** The authors have declared
that no competing interests exist.

departure, a healthcare organization loses up to $1 million in direct and indirect costs related
to lost revenue, recruitment, and onboarding costs [5]. At a national level this equates to
approximately $4.6 billion annually [6]. Professional burnout has been identified as a major
driver of physician turnover [6–9], and has been found to be strongly associated with physicians' intention to leave their current position over the next 24 months [8, 10, 11], reduce their
professional effort [12], and actually depart from the practice [8, 10, 13].

Burnout is caused by a complex interplay of multiple interdependent factors including: [1]
practice efficiency such as frustration with the electronic health records (EHR) and its poor
usability [2, 14–21] organizational leadership culture [22, 23]. Clerical burden from documentation needs [14] and overflowing inbox messages and notifications [17, 24] are causing physicians to spend an inordinate amount of time on EHR-related activities [25–27]. With an
increasing number of part-time physicians and younger physicians prioritizing work-life balance [28], practice leaders require effective strategies for recruitment and retention of talented
clinicians. Currently, practice leaders still rely on self-reported data (through physician surveys) to identify physician burnout, job satisfaction, and intention to leave [7, 11, 15, 29–31].
Given limitations of self-report, including response fatigue and bias [32], more accurate and
reliable tools for identification of physicians at-risk for departure are needed such that timely,
targeted, and appropriate interventions can alter their course and prevent departure.

EHR audit log data track how physicians spend time on EHRs [25, 33] thereby providing
an automated, objective data source that has been associated with burnout [34]. Core timebased EHR use metrics have been proposed and implemented with EHR audit log data to standardize ambulatory physician EHR use measurement normalized to 8 hours of scheduled
patient time [35, 36]. A two-year cohort study of 314 ambulatory physicians reported that physician productivity and several core EHR use metrics were associated with physician departure
[21]. Counterintuitively, less time spent on the EHR (especially inbox time) was associated
with higher rates of physician departure. In the present study, we extend this cohort through
the Covid-19 pandemic and develop a more comprehensive predictive model by adding multiple new metrics more recently found to be associated with burnout and using machine learning techniques to identify early predictors of physician departure with internal validation on
an unseen hold-out data set.

## Methods

### Study design, setting, and participants

This retrospective cohort study included training, validation, and testing arms to develop a
prediction algorithm for physician departure from the ambulatory division of a large, multispecialty practice network in the northeast region of the United States from July 2018 to May
2021. All ambulatory physicians practicing in the network were eligible. The study protocol
was approved by the practice network's institutional review board (protocol #072105) with a
waiver of informed consent since all data were de-identified and no protected health information was collected. Reporting adheres to the Strengthening the Reporting of Observational
Studies in Epidemiology (STROBE) statement checklist for cohort studies [37].

### Data sources/Variables

Monthly observations were collected for each physician participant throughout the study
period in the broad categories of physician characteristics, EHR use, clinical productivity, and
temporal trends and events. Any data on EHR use prior to the hire date and subsequent to the
departure date was not included. The primary outcome for each observed physician month
was whether departure (as determined by the physician's termination date) occurred within

the subsequent 6 months. Physicians who were involuntarily terminated were excluded from the analysis. Specific voluntary reasons for departure were not available. Data on physician characteristics were collected from the practice networks' Human Resources data records. Physician characteristics included medical specialty, tenure (time since hiring date) physician age group, and physician gender. All sites use a single instance of the Epic EHR (Epic Systems, Verona, WI). Data on physician EHR use and clinical productivity were collected from the local Epic Signal platform and Clarity database. All sites within the study had a EHR go-live date of 5 or more years prior to the start of the study. EHR use features included time on specific EHR activities (total, outside of scheduled hours, documentation, order entry, inbox, and chart review) normalized to 8 hours of scheduled patient time as defined by Sinsky et al. [35] and derived from Epic Signal per the methods described by Melnick et al. [36] as well as teamwork on orders (proportion of orders placed by other members of the care team) [35], teamwork on inbox (proportion of inbox messages responded to by other team member without physician involved), inbox message volume originating from patients (including patient medical advice and refill requests) [38], note quality manual (proportion of note characters entered by physician manually or with dictation), note quality contribution (proportion of note characters contributed by the physician) [39], and number of prescription errors (total retract and reorder errors) [40]. Clinical productivity features included patient volume (number of appointments per month), physician demand (proportion of available appointments filled) [21], work intensity (number of completed appointments per scheduled clinical hour), panel count (primary care only, number of unique patients seen in the prior two years for whom the physician was listed as Primary Care Physician in the EHR; this variable was set to *missing* for providers who were not in the specialties Internal Medicine, Family Medicine, or Pediatrics), and panel complexity (primary care only, based on Epic's average general adult risk score). To account for potential temporal trends in departure over time, features for calendar month and waves of the COVID-19 pandemic were also included. Specifically, the COVID waves used were: 0) February 2020 and before, 1) March-June 2020, 2) July-October 2020, 3) November 2020-March 2021, and 4) April-June 2021. For a given physician month, both current feature values or their longitudinal trends may be predictive of departure. To identify how longitudinal trends in features contributed to departure, exponential weighted moving averages and moving linear regression slopes were computed over the prior 90 days and added as additional features. A complete list of features and variable type is in **S1 Table**.

## Data analysis

For analyses comparing physician features, continuous variables were aggregated by taking the mean variable per-physician ignoring missing values. Statistical significance of differences within categories was assessed using Chi-Square or Fishers Exact test where appropriate. Data were split into training, validation, and testing sets with an 80%-10%-10% split using stratification to ensure similar departure rates in each set. All observations from a single physician were assigned to the same data subset to prevent information leakage.

Predicting physician departure was framed as a classification task using the primary outcome of whether departure occurred within the subsequent 6 months of an observation. Classification was performed using XGBoost (version 1.5.2) which is a computationally efficient implementation of the gradient boosted decision tree algorithm. Gradient boosted decision trees additively build an ensemble of small decision trees where each additional tree corrects errors of the prior ensemble. This technique is widely used in structured or tabular datasets on classification, regression, and longitudinal prediction problems [41–43]. Hyperparameter tuning was accomplished using 5-fold grid-search cross-validation (python package sklearn

version 1.0.1) on the training set for tree depth and L2 regularization weight. The best-fit model used binary-logistic link, depth of 6, L2 regularization weight of 40, 200 estimators, a scale positive weight of 400, area under the curve as the evaluation metric, and 10 rounds for early stopping criterion. XGBoost defaults were used for other parameters. The default methods of XGBoost were used to handle missing data, where it is considered to be a distinct value when branches are learned in training. Classification using logistic regression, gaussian naïve bayes, and random forest methods were performed for comparison. These models were fit using the python package sklearn (1.0.1).

Given the tradeoff between capturing true positives (sensitivity) and producing false positives (1-specificity) inherent in the Receiver Operating Characteristic (ROC), Youden's J index was used to determine the optimal threshold for classification [44]. Global feature significance was computed using permutation importance sampling. To examine how features contributed to risk of physician departure for each physician-month, Shapley Additive Explanations (SHAP) were used [45]. SHAP is a game-theoretic approach to model explanation which determines how each feature contributes to the overall prediction for a single observation. These contributions are additive, such that the SHAP value for each feature is the estimated log-odds contribution to the overall predicted log-odds of departure for the observation. Feature dependence plots were constructed to view contributions of single features in detail. Vertical dispersion of points in these plots were used to identify interaction effects in which the same feature value results in different SHAP values due to enhancing or attenuating interactions with other features. SHAP interaction values (which separate out the main effect of a feature from all pairwise interaction effects) were computed to examine interactions between individual variables. SHAP computations were performed using the python package shap (0.39.0). Relationships between the SHAP values and feature values were assessed for nonlinearity with the linear RESET test (python package statsmodels 0.14.0). Spearman correlation was computed to examine associations between SHAP values for a primary feature and the value of interacting features (python package scipy 1.7.3). Finally, to examine the features that were estimated to contribute the most to change in an individual physician's risk of departure during the study period, we identified physicians that had predictions of both being retained and departing within the study period. We then computed the difference in average SHAP values for feature main effects and all pairwise interaction effects for departed and retained months by physician.

All data processing and model fitting steps were performed using the Python programming language (3.8.8). Data preprocessing was performed using pandas (1.4.1). Specific packages used to perform model fitting and analyses have been referenced above. The code is available on github at https://github.com/kevkid/physician_retention_2.0.

# Results

## Participants, descriptive statistics, and outcome data

The cohort included 319 physicians from 26 medical specialties over a period of 34 months producing a total of 9928 physician month observations which were all included in the analysis. Among the 319 physicians, 132 (41.4.%) were women and 95 (29.7%) were aged 45 to 54 years. A wide range of specialties were represented in the sample including 119 (37.3%) internal medicine, 39 (12.2%) family medicine, 36 (11.3%) cardiology, and 22 additional specialties with 5% or fewer physicians. A total of 44 (13.8%) physicians departed during the study period. In comparison of physician demographic features for those who departed during the study and those who were retained, there was a significant difference across age groups (p < 0.001) with higher departure rates occurring in the 35–44 and ≥ 65 age range, there was no

**Table 1. Physician feature values stratified by departure status.**

| Feature | | Retained | Departed | p-value |
|---|---|---|---|---|
| Gender | | | | |
| | Male | 157 | 30 | 0.2 |
| | Female | 118 | 14 | |
| Age | | | | |
| | 25–34 | 8 | 2 | < 0.001 |
| | 35–44 | 54 | 12 | |
| | 45–54 | 89 | 6 | |
| | 55–64 | 77 | 5 | |
| | ≥ 65 | 45 | 19 | |
| Specialty | | | | |
| | Internal Medicine | 101 | 18 | 0.52 |
| | Family Medicine | 34 | 5 | |
| | Cardiology | 37 | 2 | |
| | Other | 103 | 19 | |
| Practice Characteristics | | | | |
| | Volume (appointments/mo) | 195 | 158 | a |
| | Demand (fraction appointments filled) | 0.75 | 0.64 | |
| | Panel Size (n) | 1145 | 927 | |
| | Panel Complexity | 1.31 | 1.82 | |
| | Intensity (completed appointments/hr) | 1.84 | 1.54 | |
| EHR use[b] | | | | a |
| | Total EHR time | 5.8 | 5.8 | |
| | Note time | 1.9 | 1.7 | |
| | Order time | 0.8 | 0.9 | |
| | Inbox time | 0.8 | 0.8 | |

[a] Mean values provided for reference. Statistical tests not performed as these features represent aggregations of both repeated measures and non-stationary values. The remainder of the analyses consider observations on the physician-month level to address.

[b] Normalized to an 8 hour work day.

significant difference in gender (p = 0.2), and there was no significant difference across specialties (**Table 1**).

## Main results

Performance of the XGBoost classification algorithm in predicting departure on the unseen testing dataset yielded an ROC Area Under the Curve (AUC) of 0.82 and an accuracy of 97% as shown in **Fig 1**. The optimal threshold for model performance determined by Youden's J index occurred with a log-odds threshold of 0.002. At this threshold, sensitivity was 0.64 and specificity was 0.79. Additional classification statistics for this threshold are shown in **Table 2**. Performance of this model exceeded that achieved by logistic regression, naïve Bayes, and random forest classifiers (**S2 Table**) leading us to choose the XGBoost model as the primary predictive tool for the study.

SHAP values (see Methods) were computed for the final model to examine the contribution of each feature to risk of departure on a physician-month level (global feature significance was assessed using permutation importance sampling and is summarized in S1 Fig). The features with highest average absolute contribution are displayed in **Fig 2** and cover a range of categories including physician characteristics, EHR use, and clinical productivity. Feature values,

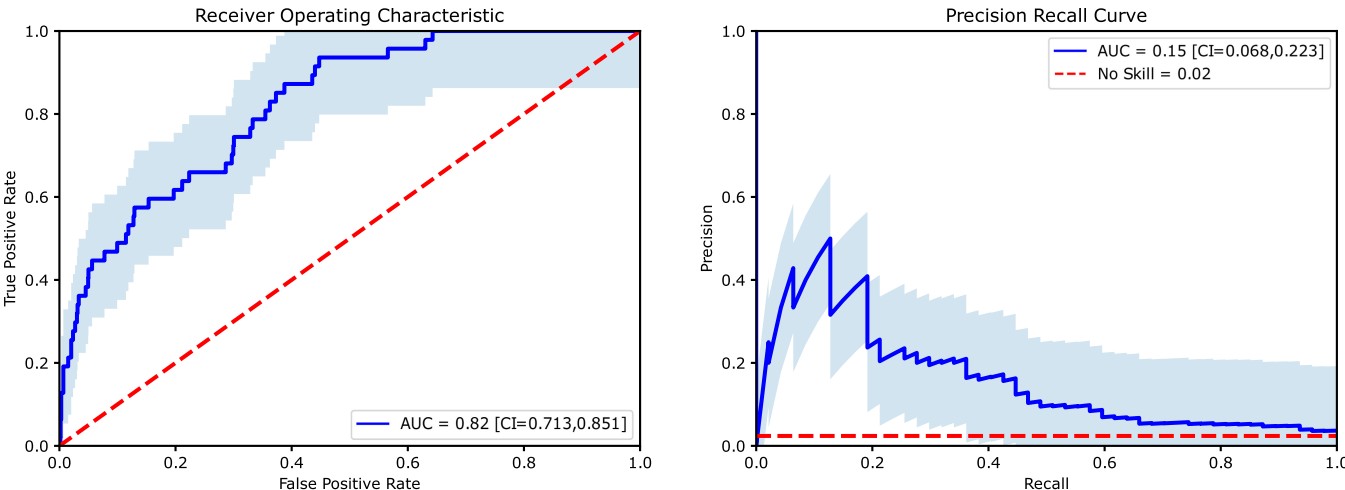

**Fig 1. Primary model performance on unseen test data.** Receiver Operating Characteristic curve and Precision Recall Curve with confidence bands and computed Area Under the Curve values. Diagonal dashed line shows the curve expected from a no-skill classifier.

encoded by point color, appear to be non-linearly related to the calculated SHAP values ($x$-axis value) suggesting complex relationships in their contribution to departure risk. Gray points denote missing values which were present in the tenure variable for 9 of the 319 physicians. Having missing tenure data was associated with increased risk of departure. Physicians with missing tenure data were more likely to be male with a trend towards older age groups (S3 Table).

Feature dependence plots for the top four features from the SHAP beeswarm plot in Fig 2 are displayed in Fig 3 to show the relationship between feature value and SHAP value, as well as possible interactions between features. Tenure contributed nonlinearly to departure risk (linear RESET test p < 0.001), with highest risk at low tenures and initially decreasing before rising again at higher tenures (Fig 3A). The vertical dispersion indicates that interactions are present. The feature with the greatest estimated interaction was the exponentially weighted average EHR time normalized to an 8-hour day. Within ordinal tenure groups by decade, high EHR times have different contributions. High EHR times tended to decrease risk of departure for physicians with tenure from 0–5 and 5–10 years (spearman correlation -0.43 and -0.50, respectively with p < 0.001), but were associated with higher risk of departure for longer tenures (spearman correlations 0.43, 0.46, 0.4, 0.28, 0.64 for 10–35 years, all p < 0.001; spearman correlations 0.17 p = 0.38, 0.02 p = 0.92, 0.40 p = 0.003, -0.31 p = 0.22). Panel complexity also showed a nonlinear contribution (linear RESET test p < 0.001) with peak risk contribution at

**Table 2. Model performance at optimal threshold using Youden's J index.** **A** 2x2 confusion matrix for the optimal threshold showing physician-month counts and **B** classification performance statistics. PPV is Positive Predictive Value; NPV is Negative Predictive Value. F1 is the harmonic mean of PPV and Sensitivity.

| A | | | | B | |
|---|---|---|---|---|---|
| | | Predicted | | Metric | Value |
| | | Retained | Departed | Sensitivity | 0.64 |
| | | | | Specificity | 0.79 |
| Ground Truth | Retained | 1523 | 397 | PPV | 0.07 |
| | | | | NPV | 0.99 |
| | Departed | 17 | 30 | F1 | 0.86 |
| | | | | Accuracy | 0.97 |

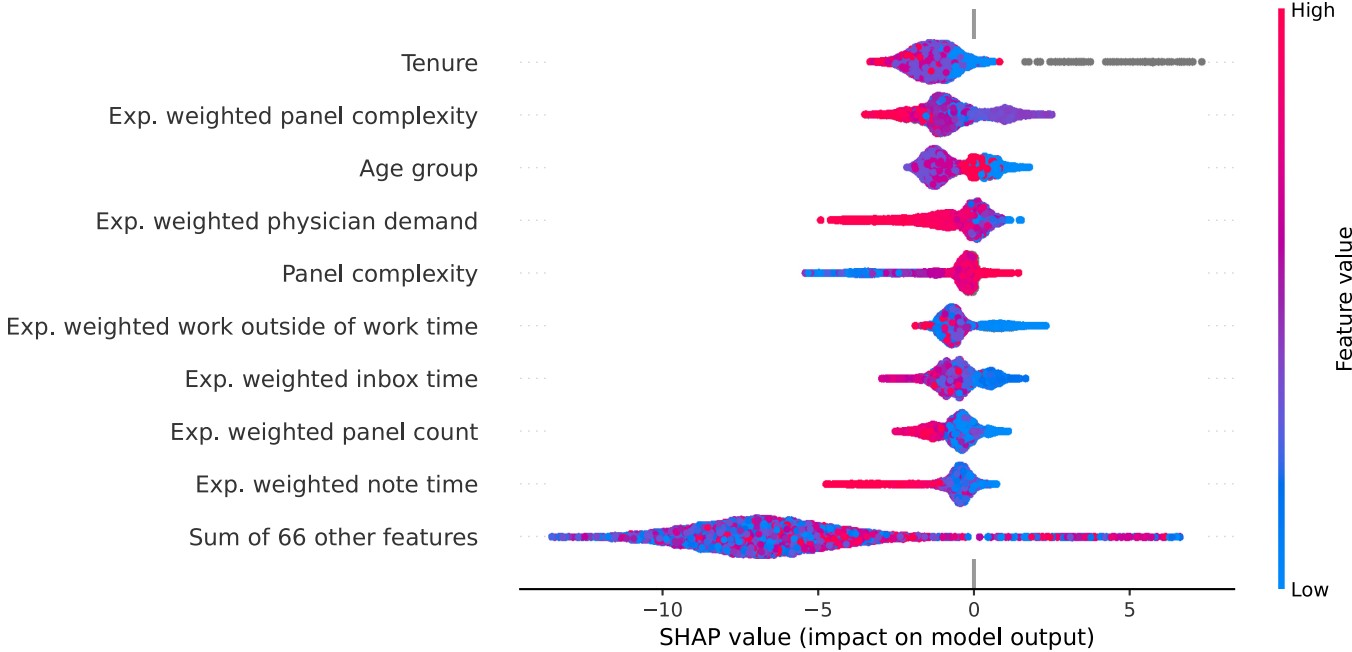

**Fig 2. Shapley Additive Explanations (SHAP) analysis Beeswarm Plot showing the 10 top features contributing to physician departure.** Each dot represents a physician-month. Positive SHAP values (right of 0.0 vertical line) indicate the feature increased the individual physician's monthly risk of departure. Actual feature values are color-coded with high feature values indicated in red, low values in blue and null values in gray.

approximately 1.7 (**Fig 3B**). The lowest and highest age groups had the highest risk of departure with protective effects estimated for most individuals aged 45 to 64 (**Fig 3C**). Interactions are evident, especially in these protective age groups. High exponentially weighted average documentation time was associated with higher SHAP values for physicians in these age groups (spearman correlation coefficient between SHAP value and interaction value 0.59 and 0.52 respectively with $p < 0.001$). Low physician demand was associated with increased risk of departure, and risk decreased nonlinearly at high demand (linear RESET test $p < 0.001$), however variation in panel count for a given level of physician demand was not readily apparent as would be present in an interaction (**Fig 3D**).

The strongest estimated interaction with tenure was the exponential weighted average of EHR use time. To examine the interaction between tenure and EHR use further, pairwise SHAP interaction values were computed between tenure and each of the exponentially weighted averages of the core EHR use metrics adapted from Sinsky et al. [35] Interaction dependence plots for tenure and the core EHR use metrics are displayed in **Fig 4**. EHR time, inbox time, and order time exhibit similar patterns. For low tenure physicians, high values meaning increased time spent on these activities are associated with the lowest risk contributions suggesting a relative protection effect. At tenures greater than 10 years, this interaction switches and high values are associated with higher risk of departure. Documentation time exhibits a different pattern. For tenures of 0–5 years, increased documentation time is associated with higher risk of departure and changes to decreased risk for 5–15 years. A complete list of spearman correlation coefficients quantifying associations between interaction SHAP values and the interaction feature value is available in **S4 Table**.

The risk of departure for many individual physicians changed throughout the study period. For example, the features that contributed to risk of departure for any one physician-month observation could be different when an individual physician transitioned from low to high risk

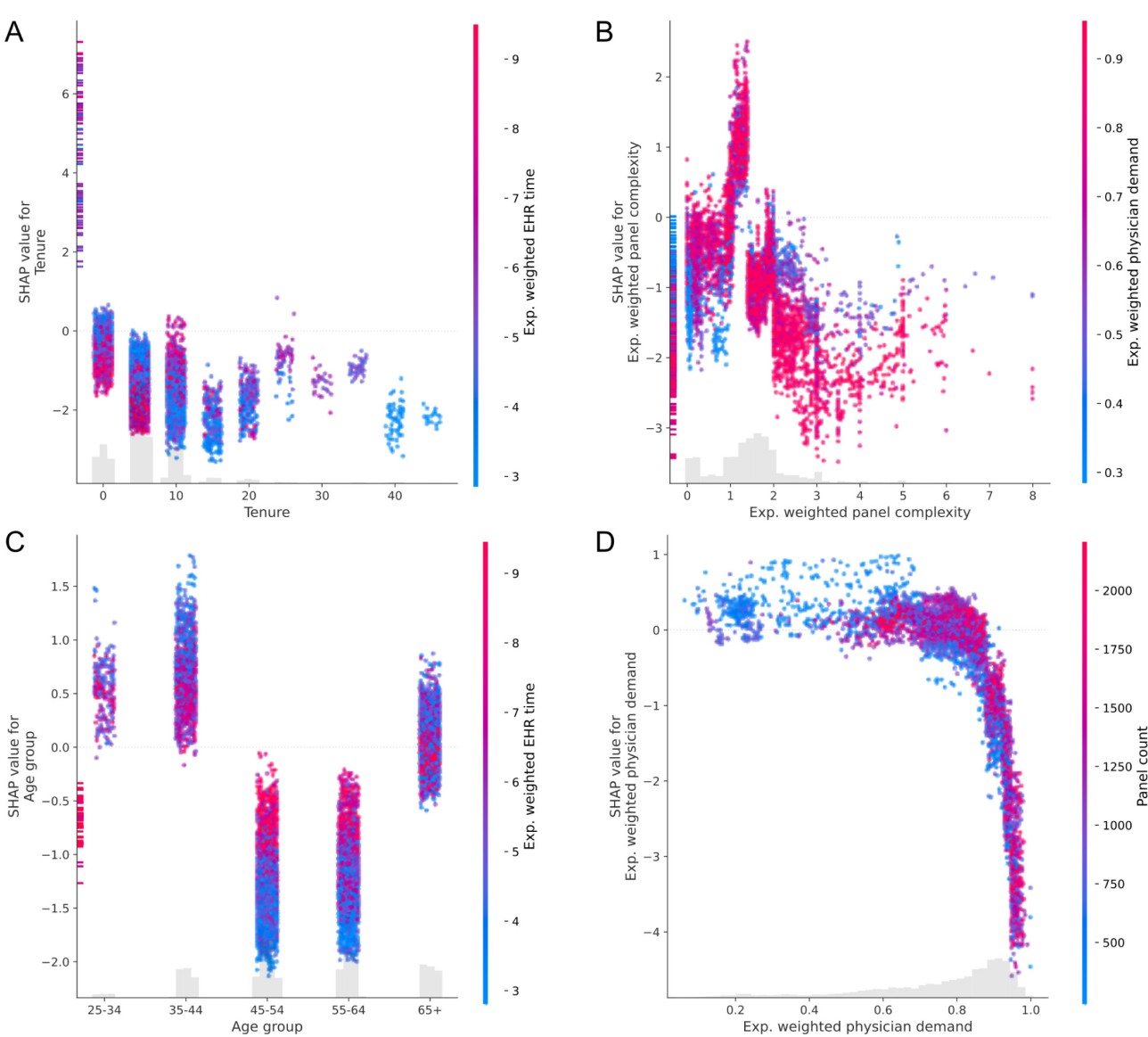

**Fig 3. Dependence plots for the features with greatest average absolute SHAP values.** Feature value is displayed on the *x*-axis and the associated SHAP value is displayed on the *y*-axis. SHAP values greater than zero denote an increase in risk. Color shows the value of the secondary feature estimated to have the strongest interaction with the primary feature.

of departure. Features with the greatest SHAP value change when physician classification changed during the study period are displayed in **Fig 5**. Interaction effects were included since the absolute value of some features (such as tenure) did not change during the study period, but the contribution of these features to risk could change due to interaction with other dynamic features. This shows which changes in main effects and interaction effects contributed the most to the change in classification status for an individual physician. The top three features with the largest average contribution to an individual physician's change in classification were the main effect of teamwork on inbox tasks, main effect of documentation time, and the slope of the panel count.

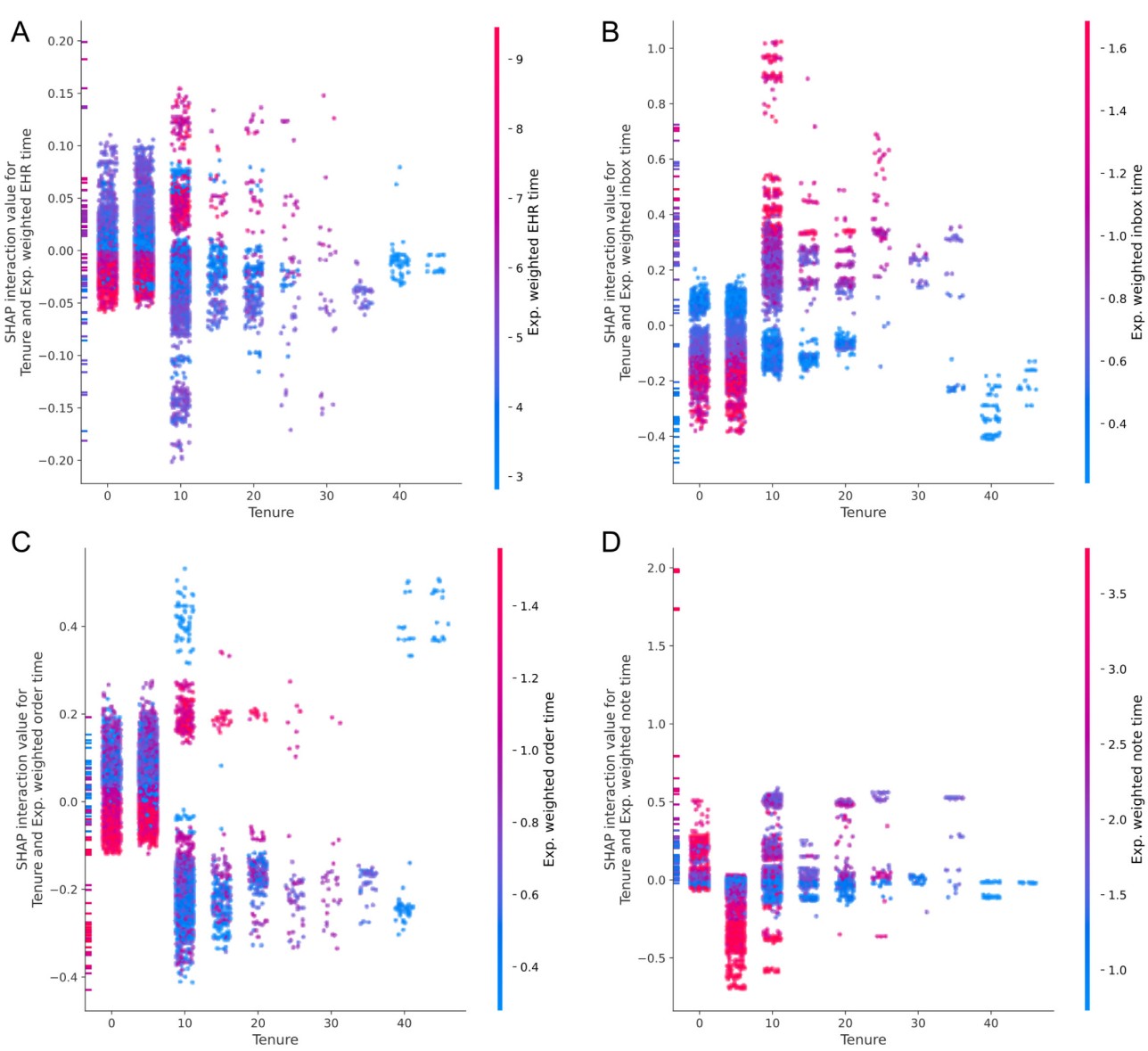

**Fig 4. SHAP values for interactions between tenure and EHR use metrics.** For each plot, the SHAP value for the interaction between tenure and an EHR use metric is shown as a function of tenure. The x-axis displays the value of tenure, the y-axis shows the SHAP value, and the color encodes the value of the EHR use metric. SHAP values for interactions above zero denote an increase in risk.

## Discussion

Although reasons for physician departure from a practice are complex, this 34-month longitudinal cohort study of 319 physicians from a large multispecialty ambulatory practice demonstrates that a machine learning model can aid in identifying those at risk of departure with a 6-month lead time with a ROC-AUC of 0.80. Using SHAP values, the predictions of this model can be deconstructed to explain the contributions of individual features to departure risk at the physician-month level. In the study population, the features with greatest average contribution to departure risk came from multiple categories including physician demographics, EHR use metrics, and patient panel characteristics. The top four contributing features

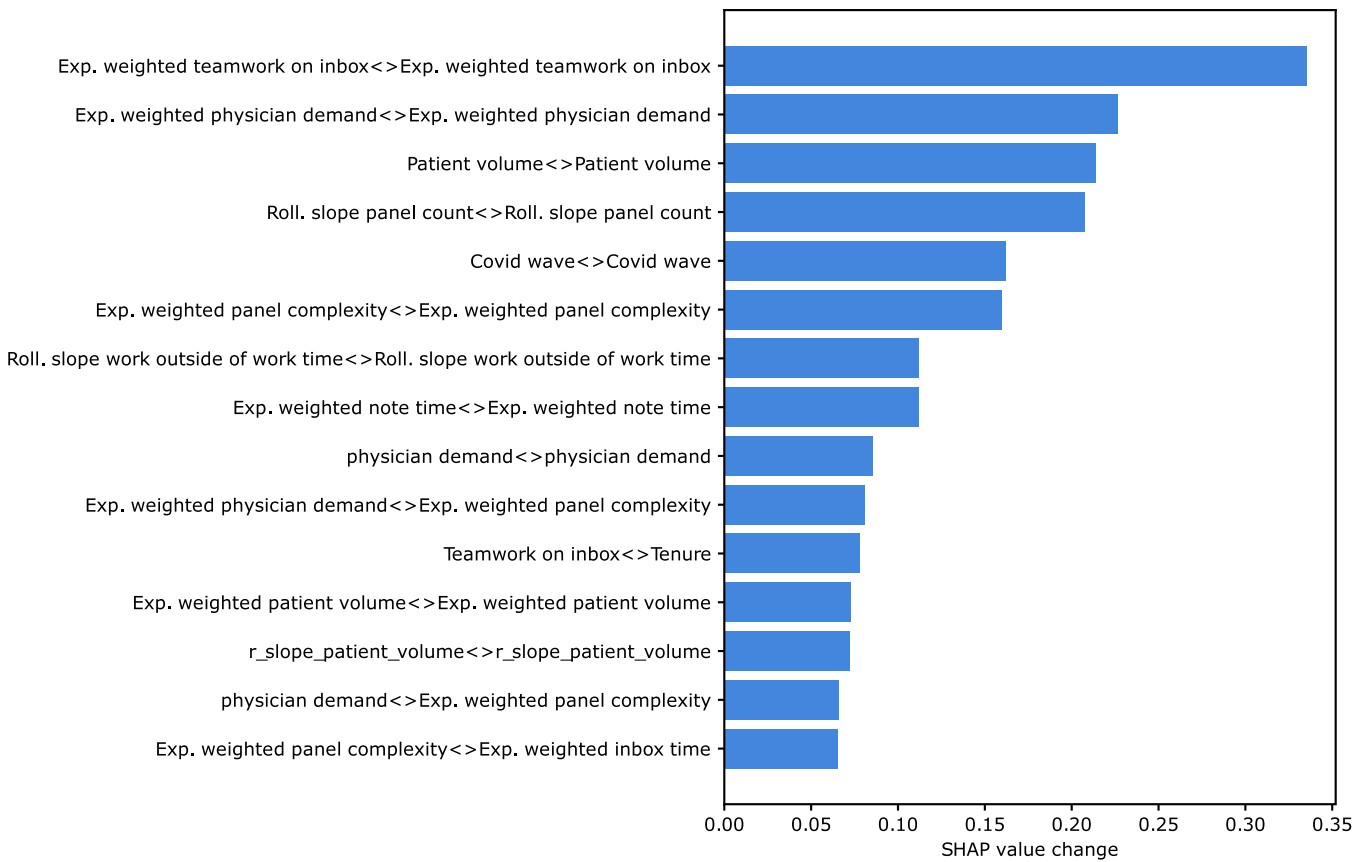

**Fig 5. Features with greatest SHAP value change when physician classification changed.** The top 15 values for average per-physician change in log-odds contribution for each feature and pairwise interaction as estimated by SHAP are displayed.

were physician tenure, physician age group, and exponentially weighted averages of panel complexity and physician demand. The contributions from individual features were non-linear with main effects moderated by interaction. For example, tenure was the feature with the highest average contribution to departure risk in the model, and computing interaction effects revealed that the risk conferred by a specific tenure is moderated by the time spent on EHR use. High use at low tenures tended to decrease risk of departure, whereas high use at higher tenures tended to increase risk. The highest contribution of tenure to departure risk came for physicians whose tenure data was missing. While certain features contributed to high absolute risk, like age and tenure duration, these were relatively static during the study period. An analysis of the features contributing to the dynamic portion of risk showed the top four features leading to changes in physician classification were teamwork on inbox tasks, note time, panel count, and panel complexity. Taken together, these findings demonstrate that machine learning algorithms, which can identify complex interaction and non-linear effects can be used to predict physician departure and provide an interpretable summary of what factors contributed to calculated risk of departure for each physician.

Machine learning models provide a flexible approach to tasks where a set of important features is hypothesized, but the direct relationships between these features and the outcome is not known and may include non-linearities and interactions. The ability to detect complex interactions is a strength of the model presented here and expands current knowledge of factors associated with physician departure [7, 21, 46]. Among the machine learning classification

algorithms tested in this study, XGBoost provided the most accurate performance and did not require imputation of missing data. These are two strengths which led to its choice as the primary predictive model for the study. Age is an example of a feature with nonlinear risk contributions that would be masked in a linear model, with high-risk contributions at low and high ages and protective effects from ages 45–64. This approach can also incorporate the status of having missing data to estimate risk. The high-risk contribution of having missing tenure data may indicate shared factors between the processes that led this data to be missing and reasons for departure. Using gradient boosted trees to resolve interactions between tenure and core EHR use, this model expands on previous counterintuitive findings of a multivariable regression model which found that low EHR use time was associated with physician departure [21]. A tenure-dependent interaction was found between tenure and total EHR, inbox, order time, and documentation time. Physicians with low tenure but high EHR utilization are at lower risk of departure than their similar-tenured peers. This predicted effect reverses, however, for increased tenure with high EHR utilization indicating higher risk than similar tenured peers. This pattern is consistent for total EHR time, inbox time, and order time. Note time departs from this trend, however with increased time associated with departure for the shortest tenure physicians. This indicates that both who and how the EHR is being used are important factors in determining risk of departure. Future qualitative research could be aimed to understand how these differences arise across tenure groups and if high EHR use is a marker or causative feature for upcoming departure. An additional strength of this model is its ability to generate dynamic monthly risk predictions for each physician. The factors contributing to changes in risk for an individual physician-month may be different from the factors with the largest overall contribution to risk. The features with highest average change in risk contribution were teamwork on inbox, documentation time, panel count, and panel complexity. Although this analysis cannot attribute causality to features impacting predicted departure risk, both teamwork on inbox and documentation time are intervenable features for which targeted changes could be designed. For example, structures for inbox task distribution or the addition of documentation assistants like dictation tools or scribes could be implemented. Panel count and complexity are both potential indicators that a practice is winding down, and although not intervenable, could provide early warning about an impending departure.

## Limitations

This study has several limitations. First, although the model provides which features contribute to risk for a given physician in a given month, this represents an associative link and not a causal link. Therefore, this study does not provide evidence that certain features led to departure or that intervening upon it would reduce departure risk—even if the model predicted that the post-intervention value was associated with decreased departure risk. Furthermore, features such as tenure, while being important to the model's output, are not intervenable. Second, prediction of rare events poses a challenge because even with high specificity, low event prevalence reduces the positive predictive value. At the optimal threshold based on Youden's J statistic, the positive predictive value is 0.07. If this model were deployed in a practice to identify physicians at a high risk of departure using this threshold, this could lead to a significant number of positive predictions. In this study population there would be approximately 110 positive predictions per month corresponding with an average of 0.8 outreaches per practice per month. There are additional variables which were not available for the analysis that may be important features in prediction of departure. One of these is practice location as certain sites may have different EHR use patterns or physician turnover rates. Practice location was not available as part of the deidentification process. Lastly, this model was trained on a data from a

specific ambulatory practice which may lead to associations that are specific to characteristics of the practice. For example, physicians with missing tenure data were scored as having high risk of departure. This measurement error was used to refine risk score but is unlikely to generalize across practices with different record accuracies. While the specific parameters in the XGBoost fit derived in this study may not generalize to other populations, the approach could be applied with tuning of parameters for other practices. We note that this is not unique to either the methods or study population here, but rather a common challenge in generalizing a model across populations.

## Conclusion

This study demonstrates how multiple streams of data routinely produced by healthcare practices can be harnessed to identify physicians at risk for departure and provide personalized insights into the factors associated with this increased risk. These results could be assembled into a dashboard for practice leaders to monitor risk in real time and with a threshold designed to meet the resources of the practice and support outreach or other interventions to physicians at highest risk of departure. This work could play a major role in leadership decisions as it could reduce recruitment and onboarding costs as well as providing more stable physician-patient relationships. Furthermore, this work could reduce physician burdens related to relocation and lost wages. Although causal links cannot be identified in this study, identifying changing features in a physician with rising risk could serve as a starting point for practice leaders to engage high risk individuals to better understand their specific needs.

## Supporting information

**S1 Table. Feature list and variable types.**
(DOCX)

**S2 Table. Model comparison.**
(DOCX)

**S3 Table. Demographics of physicians with missing tenure data.**
(DOCX)

**S4 Table. Spearman correlation coefficients for interaction SHAP plots.**
(DOCX)

**S1 Fig. Permutation importance sampling.** Permutation importance sampling was performed to test the global significance of each feature in model predictions. Four features showed statistical significance with $p < 0.05$: Tenure, Exponential Weighted panel complexity, Exponential weighted teamwork on inbox, and rolling slope of the panel count.
(DOCX)

## Author Contributions

**Conceptualization:** Hyung Paek, Edward R. Melnick, Andrew J. Loza.

**Data curation:** Kevin Lopez, Huan Li, Hyung Paek, Brian Williams, Andrew J. Loza.

**Formal analysis:** Kevin Lopez, Huan Li, Andrew J. Loza.

**Funding acquisition:** Edward R. Melnick.

**Investigation:** Kevin Lopez, Brian Williams, Bidisha Nath, Edward R. Melnick, Andrew J. Loza.

**Methodology:** Kevin Lopez, Edward R. Melnick, Andrew J. Loza.

**Project administration:** Hyung Paek, Brian Williams, Bidisha Nath, Edward R. Melnick.

**Resources:** Hyung Paek, Brian Williams, Edward R. Melnick.

**Software:** Kevin Lopez, Andrew J. Loza.

**Supervision:** Kevin Lopez, Hyung Paek, Brian Williams, Bidisha Nath, Edward R. Melnick, Andrew J. Loza.

**Validation:** Kevin Lopez, Edward R. Melnick, Andrew J. Loza.

**Visualization:** Kevin Lopez, Huan Li, Edward R. Melnick, Andrew J. Loza.

**Writing – original draft:** Kevin Lopez, Huan Li, Bidisha Nath, Andrew J. Loza.

**Writing – review & editing:** Hyung Paek, Brian Williams, Edward R. Melnick, Andrew J. Loza.

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
