## [Decision Letter · Decision Letter 0]

6 Nov 2022

PONE-D-22-27737Predicting physician departure with machine learning on EHR use patterns: A longitudinal cohort from a large multi-specialty ambulatory practicePLOS ONE

Dear Dr. Melnick,

Thank you for submitting your manuscript to PLOS ONE. After careful consideration, we feel that it has merit but does not fully meet PLOS ONE’s publication criteria as it currently stands. Therefore, we invite you to submit a revised version of the manuscript that addresses the points raised during the review process.

We look forward to receiving your revised manuscript.

Kind regards,

Sathishkumar V E

Academic Editor

PLOS ONE

Journal Requirements:

Reviewers' comments:

Reviewer's Responses to Questions

**Comments to the Author**

1. Is the manuscript technically sound, and do the data support the conclusions?

Reviewer #1: Partly

Reviewer #2: Yes

2. Has the statistical analysis been performed appropriately and rigorously? 

Reviewer #1: No

Reviewer #2: Yes

3. Have the authors made all data underlying the findings in their manuscript fully available?

Reviewer #1: No

Reviewer #2: Yes

4. Is the manuscript presented in an intelligible fashion and written in standard English?

Reviewer #1: Yes

Reviewer #2: Yes

5. Review Comments to the Author

Reviewer #1: This is a novel, sophisticated example of machine learning using EHR use data to attempt to retrospectively predict physician departure. There are a number of areas the authors did well. The authors released the algorithm’s code. Abbreviations are clear or defined at first use. The authors are clear that no causal links can be identified and explain some limitations including the low positive predictive value of the ML algorithm.

Here are significant issues with the manuscript.

Graphs

-Difficult to read, the fonts are very small, difficult to understand

-Figures 4a and 4b appear identical, also Figure 4 has 2 panels with the same labels.

Unclear physician characteristics

-Does Internal Medicine mean general medicine primary care, or everyone in Internal Medicine that is not cardiology? If so, why is only Cardiology split out? If Internal Medicine means only PCPs, then it should be relabeled and explained why only cardiology is being used.

Statistics

-There are no statistical analyses done. This coupled with a low event # gives no confidence in the results and suggests over fitting of the data. There are more features than positive outcomes. Specifically;

-The AUC has no confidence intervals

-No discussion or limitation related to only 44 positive outcomes.

-No comment with non-linear associations on possibly overfitting data or discussion of noise

-Why are some continuous variables left continuous (panel complexity, physician demand, weighted EHR time, etc., and some were converted to categorical; tenure and age.

-Why choose to discuss specifically the top four contributing features in the discussion, why not two, or eight? As no statistics are given it is not clear why these 4 were selected and whether this decision was based on a bias.

Other smaller issues

ASTRACT

#1) Not sure what “tenure” means in this non-teaching, university-based practice in the abstract. Maybe use “time since hiring” at this time, or define in the abstract

INTRODUCTION

#2) P3. “Alternatively, EHR audit log data have shown tremendous potential in tracking how physicians”

-No need for “alternatively” as not related to prior paragraph.

METHODS

#3) P1. “All ambulatory physicians practicing in the network were eligible.”

-In the abstract it noted that teaching clinicians were excluded? Why only cardiologists?

#4) No comment of EHR Go-live date. If this was different for different clinicians/sites, it should have been included, if all the same, would mention this in the methods.

#5) Data sources/variables. “Remained employed by the practice network in the broad categories of physician characteristics, HER use, clinical productivity, and temporal trends and events”

-Do not understand this sentence. I would not connect employment to the categories listed. Could this be clarified?

#6) Data sources/variables. “Reasons for departure included leaving the practice network for an outside position, leaving medicine, and retirement.”

-How was this determined? Not something that would be mandatory non-research related? Were the departing clinicians asked? Wee they consented?

#7) Data sources/variables. “Number of prescription errors (total of wrong medication, wrong patient, and wrong formulation retract and reorder errors). (41)

-This reference only measured wrong patient errors. How were wrong medication and wrong formulation errors determined? I think that this is an error and only wrong patient errors were measured if the method from ref. 41 was used.

#8) Data sources/variables. “Panel count (primary care only, number of patients seen in the prior two years),”

-First, I assume that the intent is the number of UNIQUE patients seen in the prior two years. Second, is this really a measure of panel size? A clinician seeing only acute complaints could have a high panel count despite not provide primary care. In addition, if using this definition, the sum of the panel sizes of all the PCP’s can be significantly larger that the panel size of the entire clinic. Typically, only one clinician is assigned to each patient. I am OK with the variable as it is clear, but not sure that it means or should be called panel size. Maybe call it “unique patients seen”.

#9) There is no discussion on different sites. Did all the clinicians come from the same geographic site? If so, would mention this. If not, why wasn’t site a variable?

RESULTS

Main results.

#10) Performance on the unseen testing dataset yielded an ROC Area Under the Curve (AUC) of 0.80

-This study was somewhat underpowered as only 44 clinicians departed. As such, it would be nice to see the CI of the AUC to get an idea how much uncertainty the somewhat small sample is producing.

Table 1

#11) F1 is not defined

Paragraph 2

#12) While 3a is specifically discussed, it in not clearly stated where 3b, 3c and 3d are being talking about.

#13) From a statistical point of view, this text talks about non-linear changes in the risk of departure, such as high then down, then back up for instance. The outcome only had 44 events. Does this data set have the ability to statistically find these non-linear changes in the in all these variables. It is claimed in Figure 2 that these are 67 features. Is there really enough power to suggest that these non-linear relationships with the outcome are significantly better than linear, or just noise?

#14) There was also no discussion about which variables were actually statistically significantly associated with the outcome? If decision tree analysis with Gradient Boost is unable to test if a feature is statistically associated with an outcome, this needs to be explained in the methods or results. If this method cannot produce statistically analysis, would consider running a secondary logistic regression to at least show which features are statistically associated with the outcome

#15) In results page 5 (paragraph 3) says ages 35-54 are protective from departure when in discussion page 7 (paragraph 2) says ages 45-64 are protective which is supported by Figure 3c.

#16) Figures 4 has 2 panels that are labelled “Tenure and Exp. weighted EHR time”?

Discussion

#17) P1: “The highest contribution of tenure to departure risk came for physicians whose tenure data was missing.”

-This is a limitation of the data. If the method of using this missingness as a predictor were to be recommended, it would not be useful as any missing tenures could be found. What would be more interesting would be to try to figure out if the missingness had a bias? Maybe older physicians? Younger?

Reviewer #2: 1.Introduction section needs to be re-written to improve its quality and readability.

2.What is the motivation of the proposed work? Research gaps, objectives of the proposed work should be clearly justified

3.Summarize the dataset used in table (Features, Continuous/Categorical, Sample values)

4.The literature has to be strongly updated with some relevant and recent papers focused on the fields dealt with in the manuscript.

5.Authors used only XGBoost while there are several Machine learning algorithms such a SVM, KNN, GBM, LR, Random forest. Authors need to compare the results with these state of art algorithms

6. Explain why the current method was selected for the study, its importance and compare with traditional methods.

7.Authors are suggested to include more discussion on the results and also include some explanation regarding the justification to support why the proposed method is better in comparison towards other methods

8.Does this kind of study have never attempted before? Justify this statement and give an appropriate explanation to do so in this paper.

9.Quality of figures is so important too. Please provide some high-resolution figures. Some figures have a poor resolution.

10.Add Exploratory data analysis section discussing the dataset.

11. Authors mentioned "Several variables significantly contributed to predicting physician departure". How authors concluded that?. Provide valid proof for this claim. Provide results with and without these variables.

12. What kind of hyperaparameter selection strategy is used?

13. Overall writing, results and presentation needs improvement

14. Try using feature selection algorithms and feature importance representation.

15,

6. PLOS authors have the option to publish the peer review history of their article (what does this mean?). If published, this will include your full peer review and any attached files.

Reviewer #1: No

Reviewer #2: **Yes: **Usha Moorthy

---

## [Author Response · Author response to Decision Letter 0]

21 Dec 2022

A detailed point-by-point response to reviewers has been uploaded as part of this resubmission.

---

## [Decision Letter · Decision Letter 1]

27 Dec 2022

Predicting physician departure with machine learning on EHR use patterns: A longitudinal cohort from a large multi-specialty ambulatory practice

PONE-D-22-27737R1

Dear Dr. Melnick,

We’re pleased to inform you that your manuscript has been judged scientifically suitable for publication and will be formally accepted for publication once it meets all outstanding technical requirements.

Kind regards,

Sathishkumar V E

Academic Editor

PLOS ONE

Additional Editor Comments (optional):

Reviewers' comments:

Reviewer's Responses to Questions

**Comments to the Author**

1. If the authors have adequately addressed your comments raised in a previous round of review and you feel that this manuscript is now acceptable for publication, you may indicate that here to bypass the “Comments to the Author” section, enter your conflict of interest statement in the “Confidential to Editor” section, and submit your "Accept" recommendation.

Reviewer #2: (No Response)

2. Is the manuscript technically sound, and do the data support the conclusions?

Reviewer #2: (No Response)

3. Has the statistical analysis been performed appropriately and rigorously? 

Reviewer #2: (No Response)

4. Have the authors made all data underlying the findings in their manuscript fully available?

Reviewer #2: (No Response)

5. Is the manuscript presented in an intelligible fashion and written in standard English?

Reviewer #2: (No Response)

6. Review Comments to the Author

Reviewer #2: (No Response)

7. PLOS authors have the option to publish the peer review history of their article (what does this mean?). If published, this will include your full peer review and any attached files.

Reviewer #2: **Yes: **Usha Moorthy

<quillbot-extension-portal></quillbot-extension-portal>

---

## [Editor Report · Acceptance letter]

5 Jan 2023

PONE-D-22-27737R1 

­­Predicting physician departure with machine learning on EHR use patterns: A longitudinal cohort from a large multi-specialty ambulatory practice 

Dear Dr. Melnick:

I'm pleased to inform you that your manuscript has been deemed suitable for publication in PLOS ONE. Congratulations! Your manuscript is now with our production department. 

Kind regards, 

on behalf of

Dr. Sathishkumar V E 

Academic Editor

PLOS ONE